# Comparison of Short- versus Long-Course Antimicrobial Therapy of Uncomplicated Bacterial Pneumonia in Dogs: A Double-Blinded, Placebo-Controlled Pilot Study

**DOI:** 10.3390/ani11113096

**Published:** 2021-10-29

**Authors:** Aida I. Vientós-Plotts, Isabelle Masseau, Carol R. Reinero

**Affiliations:** 1College of Veterinary Medicine, University of Missouri, Columbia, MO 65211, USA; vientosplottsai@missouri.edu; 2Department of Sciences Cliniques, Faculté de Médecine Vétérinaire, Université de Montréal, St-Hyacinthe, Montreal, QC H3T 1J4, Canada; isabelle.masseau@umontreal.ca

**Keywords:** community-acquired pneumonia, aspiration pneumonia, global radiographic score, thoracic imaging, bacterial culture

## Abstract

**Simple Summary:**

Dogs diagnosed with bacterial pneumonia are often treated with long courses of antibiotics (3–6 weeks) and chest X-rays are used to help guide the duration of treatment. This is in stark contrast with humans with the same disease who are often treated for 5–10 days, and chest X-rays are not considered to be useful for monitoring response to treatment. The main goal of this study was to determine whether a shorter course of antibiotics (10 days) would be sufficient to treat dogs with bacterial pneumonia. Eight dogs with pneumonia were randomly assigned to receive 10 or 21 days of antibiotics. They were evaluated at 10, 30 and 60 days after diagnosis. At 10 days, 6/8 dogs had resolution of both clinical signs and evidence of inflammation on bloodwork and 5/8 dogs showed improvement in the chest X-rays. After 60 days, none of the dogs had clinical signs or evidence of inflammation on bloodwork regardless of antibiotic therapy duration. However, 3/8 dogs showed changes in the chest X-rays. This study suggests that a 10-day course of antibiotics may be sufficient to treat dogs with bacterial pneumonia, and chest X-rays may not be a reliable marker to monitor response to therapy.

**Abstract:**

Current treatment for canine bacterial pneumonia relies on protracted courses of antimicrobials (3–6 weeks or more) with recommendations to continue for 1–2 weeks past resolution of all clinical and thoracic radiographic abnormalities. However, in humans, bacterial pneumonia is often treated with 5–10-day courses of antimicrobials, and thoracic radiographs are not considered useful to guide therapeutic duration. The primary study objective was to determine whether a short course of antimicrobials would be sufficient to treat canine bacterial pneumonia. Eight dogs with uncomplicated bacterial pneumonia were enrolled in this randomized, double-blinded, placebo-controlled study comparing clinical and radiographic resolution with differing durations of antimicrobial therapy. Dogs received a course of antimicrobials lasting 10 (A10) or 21 (A21) days. Dogs randomized to the A10 group received placebo for 11 days following antimicrobial therapy. Patients were evaluated at presentation and 10, 30 and 60 days after the initiation of antimicrobials. At 10 days, 6/8 dogs had resolution of both clinical signs and inflammatory leukogram, and 5/8 dogs had improved global radiographic scores. After 60 days, clinical and hematologic resolution of pneumonia was noted in all dogs regardless of antimicrobial therapy duration; however, 3/8 dogs had persistent radiographic lesions. Thoracic radiographs do not appear to be a reliable marker to guide antimicrobial therapy in canine bacterial pneumonia as radiographic lesions may lag or persist despite clinical cure. This pilot study suggests a 10-day course of antimicrobials may be sufficient to treat uncomplicated canine bacterial pneumonia.

## 1. Introduction

Bacterial pneumonia is defined as inflammation of the lung due to pulmonary parenchymal bacterial infection [1]. Reportedly, the two most common types of bacterial pneumonia in dogs are community-acquired pneumonia (CAP) and aspiration pneumonia (AP) [2]. Community-acquired pneumonia develops after close contact with another dog harboring a contagious bacterial pathogen; additionally, contagious viral pathogens can predispose to secondary bacterial pneumonia with infection by local bacteria. Aspiration pneumonia, with the aspiration of bacteria from the upper aerodigestive tracts or with sufficient damage to mucosal defenses to allow a secondary bacterial infection, can lead to secondary bacterial pneumonia [3]. However, despite the widespread use of antimicrobials for aspiration pneumonia, secondary bacterial infection does not always occur as the aspiration of oropharyngeal or gastric materials can cause a sterile chemical pneumonitis [4]. Other causes of bacterial pneumonia include ventilator-associated pneumonia (VAP) and pneumonia secondary to local or systemic immune defects or immunosuppression [5].

Assuming bacterial infection, the treatment of pneumonia heavily relies on antimicrobials. These should ideally be based on culture and sensitivity as CAP may primarily be viral and AP may be non-infectious inflammatory, and because antimicrobial resistance patterns may be difficult to predict. In dogs strongly suspected of having bacterial pneumonia, initial empiric antimicrobials covering the most common bacteria are chosen—ideally while sensitivity results are pending. In a study evaluating bronchoalveolar lavage fluid (BALF) cultures from 502 dogs with respiratory disease, enrofloxacin had the best pattern of susceptibility overall (86% of all isolates and 87% of Gram-negative isolates) and amoxicillin/clavulanic acid had the best susceptibility pattern in Gram-positive bacteria (92% being susceptible) [6]. In another study, 29/111 (26%) of dogs treated for bacterial pneumonia with empirically selected antimicrobials had at least one bacterial isolate that was resistant to the antimicrobial selected. In dogs with a history of antimicrobial administration within the preceding 4 weeks, the incidence of antimicrobial resistance was even higher (54%), leading to the suggestion that airway sampling for bacterial culture and susceptibility may be helpful in guiding antimicrobial therapy [2].

Following empiric antimicrobial use, subsequent de-escalation based on sensitivity is recommended to reduce bacterial resistance selection pressure as well as treatment costs [7]. The standard of practice in veterinary medicine is to treat dogs with bacterial pneumonia for a minimum of 3–6 weeks and at least 1–2 weeks after the resolution of clinical and radiographic signs of pneumonia [5]. There are no studies in the veterinary literature (retrospective or prospective) evaluating the optimal length of therapy or objective measures to guide the duration of therapy in dogs with culture-documented bacterial pneumonia.

Overuse and inappropriate use of antimicrobials in humans and in food animals are well-known contributors to antimicrobial resistance and thus constitute a serious threat to public health [8] Antimicrobial resistance is also highly relevant in companion animal health. Although there are limited data evaluating the impact of antimicrobial resistance in veterinary patients, antimicrobial resistance is also highly relevant for canine health. Similar to humans, it is likely that multidrug-resistant infections can be associated with longer hospitalization time, greater economic impact and potentially even poorer outcomes [9]. The need to improve guidelines related to the use of antimicrobials in companion animals is particularly relevant since they are in close contact with humans and may share or amplify resistant bacteria or act as reservoirs for their human owners [10,11]. All users of antimicrobial drugs, including veterinarians, have a responsibility to use them judiciously and prudently. While this has yet to gain traction for the treatment of canine bacterial pneumonia, an urgent need to implement antimicrobial stewardship programs in companion animals has been recognized and an American Veterinary Medical Association Task Force for Antimicrobial Stewardship in Companion Animal Practice has been formed [11,12,13]. It is highly likely that the current standard of care using protracted courses (3–6 weeks or more) of antimicrobials for uncomplicated canine bacterial pneumonia opposes guidelines for good antimicrobial stewardship and may be a harmful practice.

In their guidelines for antimicrobial use in the treatment of canine bacterial pneumonia, the International Society for Companion Animal Infectious Disease stated insufficient data supporting durations of treatment shorter than 4–6 weeks [7]. While the recommendation for the re-evaluation of patients based on clinical, hematologic, and radiographic findings is no later than 10–14 days after the initiation of therapy, they acknowledged the need for additional studies evaluating shorter treatment duration [7]. To address the shortcoming in the peer-reviewed literature, this pilot prospective, double-blinded placebo-controlled trial compared outcome parameters for the clinical and hematological cure of uncomplicated bacterial pneumonia after receiving antimicrobials for 10 or 21 days. Dogs with uncomplicated bacterial pneumonia, defined as cases of CAP, AP (in which bacteria could be cultured from lavage fluid) or VAP, were selected for this study. To prevent challenges in distinguishing a single infection from persistent (due to immune compromise) or recurrent bacterial pneumonia during the study period, dogs initially diagnosed with any condition predisposing them to recurrent infections were excluded. Additionally, since the current standard of care involves monitoring by radiographic resolution, another goal was to determine whether serial radiographic examinations correlated well with clinical and hematological resolution of canine bacterial pneumonia and may therefore be used to guide antimicrobial therapy.

## 2. Materials and Methods

### 2.1. Animals

Client-owned dogs presented at the Veterinary Health Center, University of Missouri from April 2017 to July 2018 and suspected of having bacterial pneumonia and undergoing airway lavage were screened for enrollment in this double blinded, placebo-controlled study. Study inclusion mandated the collection of bronchoalveolar lavage fluid (BALF) for culture and sensitivity to definitively document bacterial infection. Dogs were prospectively enrolled if diagnosed with uncomplicated bacterial pneumonia (i.e., CAP, secondary to a sole episode of aspiration in which cultivable bacteria were identified, or VAP in which the ventilator was used for acute support in absence of chronic respiratory disease). Bacterial pneumonia was documented with septic suppurative inflammation on BALF with either intracellular bacteria observed on cytologic evaluation or a positive BALF culture. Informed client consent was obtained, and the study was performed according to institutional guidelines for animal care and use (IACUC protocol #8942). Other inclusion criteria were baseline complete blood count (CBC) and three view thoracic radiography acquired at presentation. Dogs were excluded if they received a single dose of cefovecin or >1 dose of other antimicrobials in the preceding 14 days, or if they had an underlying condition predisposing them to recurrent pneumonia (e.g., recurrent aspiration from dysphagia; primary or secondary immunodeficiency or immunosuppression; foreign body pneumonia; or lower respiratory anatomic disorders such as bronchiectasis or bronchomalacia).

### 2.2. Sample Collection

Anesthetic protocols for BALF collection were tailored to individual patient needs at the discretion of a board-certified anesthesiologist. Bronchoalveolar lavage fluid was collected by instilling a 20 mL aliquot of warmed sterile saline through the biopsy channel of a sterilized endoscope. A minimum of 2 mL of BALF was submitted for cytologic analysis that included a nucleated cell count, cytocentrifugation and, if sample quality allowed, a cell differential count. A minimum of 1 mL of BALF was submitted to the University of Missouri Veterinary Medical Diagnostic Laboratory for aerobic and anaerobic culture. For aerobic culture, aliquots of BALF samples were added to the blood agar and MacConkey agar plates and incubated at 35 °C. For anaerobic culture, aliquots of BALF samples were added to chocolate agar plates and incubated at 35 °C with 95% air and 5% CO_2_. All cultures were incubated for 24–36 h. Bacterial isolates were Gram stained and identified with the Automated Sensititre AP-80 or AP-90 for aerobic bacteria or the MALDI-TOF identification system (Matrix-Assisted Laser Desorption/Ionization–Time of Flight: Bruker Daltonics, Inc. 40 Manning Road, Manning Park, Billerica, MA, USA) or conventional biochemical reactions [14]. Aerobic susceptibility testing was performed using the Sensititre Micro-Broth (Thermofisher Scientific 12076 Santa Fe Drive, Lenexa, KS, USA) dilution minimal inhibitory concentration system.

### 2.3. Treatment Groups

Dogs were randomized via a table of random numbers with a 1:1 enrollment into a short 10-day (group A10) or long 21-day (group A21) course treatment. Both groups were treated with IV antimicrobials (ampicillin/sulbactam 30 mg/kg IV every 8 h and enrofloxacin 15 mg/kg IV every 24 h) while hospitalized. Once patients were able to receive oral medications or were discharged, they were transitioned to oral amoxicillin/clavulanic acid (20 mg/kg PO every 12 h) and enrofloxacin (15 mg PO every 24 h) for the remaining additional days for a total of 10 or 21 days. Dogs in group A10 received the placebo for 11 days to bring the total time of treatment to 21 days in each group. To assist with blinding, each dog received two containers of oral medication for the initial days (antimicrobials) and the subsequent 11 days (antimicrobials (A21) or placebo (A10). As soon as antimicrobial sensitivity panels were available, therapy was de-escalated or altered to address just the relevant pathogen with placebo still administered as outlined above. All other aspects of case management such as the administration of IV fluids, supplementary oxygen, and implementation of nebulization and coupage were left to the discretion of the attending clinician.

### 2.4. Follow-Up Evaluations and Data Analysis

Follow-up evaluations included a complete physical exam, CBC and thoracic radiography and were scheduled at days 10 (range, 10–13), 30 (range, 25–35) and 60 (range, 55–65) from the start of antimicrobial therapy. Outcome parameters included clinical cure (resolution of fever, cough, tachypnea, and respiratory distress), hematologic cure (resolution of segmented neutrophilia, band neutrophilia or neutrophil toxicity) and radiographic resolution. All radiographic studies were reviewed, disabling any annotation on the image, in a single session by a board-certified diagnostic imaging specialist (IM). Several steps were first taken to perform the radiographic evaluation without awareness of the patient case number and acquisition date. First, a list of 32 6-digit random numbers was generated using a free web software (https://www.graphpad.com/quickcalcs/randomN1.cfm, accessed on 8 November 2019). After importing radiographic studies into an individual album using OS X operated 64-bit image viewer (Horos, v3.3.5, The Horos Project.org), each study was anonymized for patient name, age, birth date and acquisition date and the patient ID was replaced by the random number generated. Random numbers were not duplicated and were used in the order listed. Then, studies were sorted by patient ID (i.e., generated random number). At the end of the radiographic assessment, results for each study were matched to the original patient information and appropriate acquisition date. Each study consisting of recumbent right lateral, left lateral and ventrodorsal radiographic projections was assessed for lesions and assigned a global radiographic score (GRS) according to a severity scoring system used for the assessment of pneumonia in children and adapted from Taylor et al. [15] (Table 1).

### 2.5. Statistical Analysis

Where appropriate, data were descriptively presented. Statistical analysis was performed using commercially available software (SigmaPlot, Systat Software Inc., Chicago, IL, USA). A Mann–Whitney rank sum test was used to compare variables such as age and length of hospitalization between groups. A two-way repeated-measures ANOVA was used to compare differences in GRS at each time point within each group and at each time point between groups (A10 or A21). A pairwise multiple comparison was performed using a Bonferroni *t*-test. Significance was set at *p* < 0.05.

## 3. Results

Twenty-three dogs with uncomplicated bacterial pneumonia having BALF collection were screened for study inclusion with eight dogs ultimately enrolled (four in the A10 group and four in the A21 group; Figure 1). Dogs in the A10 group had a mean age of 6.4 years, including three females spayed, one male intact, and one of each of the following breeds: Golden Retriever, Miniature Schnauzer, Labrador retriever and a mixed breed dog. Dogs in the A21 group had a mean age of 5.5 years, including two female spayed, one female intact and one male castrated; two Yorkshire Terriers, a German Shepherd and a Pomeranian. There was no significant difference in age between groups (*p* = 0.886). The initial clinical signs, physical examination findings, global radiographic score, BALF analysis and final diagnoses are summarized in Table 2. At least one of the following hematologic abnormalities was detected in all dogs: segmented neutrophilia (*n* = 6), band neutrophilia (2) and neutrophil toxicity (3). Ancillary therapies, antimicrobial treatments and de-escalation are summarized in Table 3. There were no reported complications associated with the BALF collection. 

A variety of ancillary therapies were employed. All hospitalized dogs were nebulized with sterile saline and coupaged. Seven dogs received IV fluid therapy and one dog received mechanical ventilation due to perceived impending respiratory exhaustion from suspected non-cardiogenic pulmonary edema—this dog developed VAP. Dogs in this study were hospitalized for an average of 3.5 days (range 0–7). There was no significant difference in length of hospitalization between treatment groups (*p* = 0.11).

### Outcomes

Outcome parameters (clinical cure, hematologic cure, and GRS) were divided into treatment groups, displayed over the course of the study, are summarized in Table 4. 

Clinical cure: By the time of the first follow-up evaluation, all dogs had normal body temperature, respiratory rate, and effort, and 6/8 dogs had complete resolution of cough. For these six dogs, no further coughing was reported throughout the remainder of the study. One of the two dogs in which the cough was persistent had concurrent respiratory distress which was resolved within 36 h of hospitalization; the persistent chronic cough was attributed to the comorbid condition of tracheal collapse. An additional dog presented with a moist cough and anorexia. The anorexia and moist cough were resolved; however, the dog had a mild intermittent dry cough at the time of the first follow-up visit. The moist cough recurred by the second follow-up visit but resolved between the second and third visit with recommendations for nebulization and coupage, and no further antimicrobial therapy. At the time of the third visit, owners reported that the cough character had changed back to an intermittent dry cough. This patient was later diagnosed with laryngeal paralysis.

Hematologic cure: At the time of the first follow-up evaluation, the band neutrophilia and neutrophil toxicity resolved in all except one dog which had a mild, persistent band neutrophilia of 0.699 × 10^3^/uL (0.000–0.260). No band neutrophilia nor neutrophil toxicity was detected during subsequent visits in any of the dogs. While some dogs had a modest segmented neutrophilia present during the second and third follow-up visits, they also had a lymphopenia, consistent with a stress leukogram [16].

Radiographic resolution: Based on the GRS there was no significant difference in the severity of radiographic lesions at each of the time points between the two treatment groups (*p* > 0.05). Overall, there was a significant decrease in the GRS over time regardless of the group (*p* < 0.0001). In the A10 group, there was a significant overall decrease in the GRS at each follow-up visit compared to baseline (*p* < 0.05). However, one of the dogs had a resolution of radiographic findings at the first visit (GRS = 1), and despite lacking an identified predisposing condition, there was evidence of re-aspiration at the second visit (GRS = 3). This dog did not have a second BAL but was treated with nebulization and coupage without antimicrobials and on the third visit had a GRS of 2. In the A21 group, there was no significant difference in the GRS between the baseline and the first follow-up visit (*p* = 0.532), but there was a significant decrease at the second (*p* = 0.012) and third visit (*p* = 0.003). Despite clinical and hematologic resolution of pneumonia, by the final study visit at day 60, 3/8 dogs had persistent radiographic lesions; two dogs in the A10 group had a final GRS of 2, and one dog in the A21 group had a final GRS of 3. Figure 2 provides an example of radiographic resolution of pneumonia.

## 4. Discussion

In the current prospective, a double-blinded, placebo-controlled study evaluating dogs with a definitive documentation of bacterial pneumonia using BALF analysis, there was no significant difference in clinical or hematologic cure when comparing 10 or 21 days’ duration of antimicrobial therapy. Thus, this pilot study suggests that the treatment of uncomplicated canine bacterial pneumonia does not require the protracted use of antimicrobials. This may be particularly important considering the empiric overuse of antimicrobials for Canine Infectious Respiratory Disease Complex (i.e., CAP) and AP in the absence of bacterial infection confirmed via BALF analysis and culture. Furthermore, as shown in humans with bacterial pneumonia, thoracic radiography does not appear to accurately reflect a positive response to therapy [17]. All dogs achieved clinical and hematological cure of bacterial pneumonia with 38% (3/8) dogs having persistent thoracic radiographic lesions at 60 days. Recently published guidelines for the use of antimicrobials for the treatment of respiratory infections in dogs recognized the lack of evidence for traditional long courses of antimicrobials. These guidelines recommended that dogs diagnosed with bacterial pneumonia should be prescribed antimicrobials and re-evaluated 10–14 days after initiating treatment, and at that time, decisions to continue therapy should be based on clinical, hematological, and radiographic findings [7]. Collectively, results from our study suggest that a short course (10 days) of antimicrobials may represent an appropriate therapy for canine bacterial pneumonia and that persistent radiographic lesions should not be used to increase the duration of antimicrobial therapy in the presence of clinical and hematologic cure.

Veterinary textbooks have long perpetuated a management protocol for canine bacterial pneumonia using 3–6 or more weeks of antimicrobials, without this recommendation being evidence-based. To date, there have been no peer-reviewed studies requiring BAL-confirmed bacterial infection evaluating duration and response to antimicrobial treatment or metrics to assess resolution in dogs with bacterial pneumonia. A prospective observational study compared the outcomes of dogs with “presumptive” (i.e., not culture confirmed) bacterial uncomplicated pneumonia (CAP and AP) treated with a short (≤14 days) or long (>14 days) course of antimicrobials [18]. No significant difference in radiographic resolution or relapse rate at 3 months was noted between dogs treated with short and long antimicrobial courses. However, the preliminary data presented in that study had several important limitations. First, the lack of randomization and standardization of treatment could have allowed for the introduction of investigator and owner bias. Second, dogs of that study lacked diagnostics to definitively determine whether cases of aspiration pneumonia had a secondary bacterial infection, with only 3/47 (6%) dogs having either trans- or endotracheal washes [18]. As AP may not necessarily be associated with bacterial infection, the apparent response to administered antimicrobial therapy is misleading. Finally, the definition of relapse (noted to be 17%) was vague and defined as lack of resolution of radiographic or clinical signs, owner-reported respiratory signs consistent with pneumonia, or re-presentation to a hospital for evaluation of respiratory signs within 3 months of initial presentation. There were no diagnostic tests performed to determine whether those patients had in fact relapsed or developed other respiratory diseases such as tracheal collapse, etc. 

The optimal duration of antimicrobial therapy for canine bacterial pneumonia is unknown. In humans, one meta-analysis of randomized controlled trials reported similar outcomes in patients with CAP receiving antimicrobial therapy for 3–7 days compared to longer courses [19]. Of note, guidelines for antimicrobial duration in humans with sepsis, severe sepsis and septic shock are only 7–10 days [20]. Other meta-analyses of randomized controlled clinical trials in humans with bacterial pneumonia reported no significant difference between short and long courses of antimicrobials with respect to clinical cure, bacterial eradication, adverse effects, or mortality [21]. This pilot study in dogs with BAL-confirmed bacterial pneumonia demonstrated parallel results to what has been reported in humans regarding a short length of antimicrobial use. 

Monitoring the resolution of bacterial pneumonia in dogs has traditionally been heavily relied on serial thoracic radiography. However, this practice is not based on scientific data from the peer-reviewed literature, and as suggested in this study, is a potentially misleading practice. In humans, radiographic improvement lags the clinical response. Furthermore, it offers no benefit in identifying treatment failure over physician or patient assessment of symptoms, underscoring the lack of utility of this diagnostic to guide therapy [17,22,23]. In one study of human CAP comparing short and long courses of antimicrobials, radiographic persistence of lesions occurred in 58% of patients at day 10 and 32% at day 28 [16]. Clinical cures were perceived in approximately 90% of patients at these two time points. The study concluded that routine chest radiographs should not be used to monitor the resolution of CAP in people. Based on the results of our study, it may be more prudent to perform radiography in dogs with persistent clinical or hematologic abnormalities (e.g., perhaps at 10–14 days after initiation of antimicrobials) or in those with initial improvement suffering a relapse rather than repeating thoracic radiography at pre-specified time points after starting antimicrobials in dogs with bacterial pneumonia [7]. In cases of persistent clinical signs or evidence of systemic inflammation, further diagnostics such as respiratory fluoroscopy, videofluoroscopic swallow study, functional laryngeal exam, thoracic computed tomography, bronchoscopy, and bronchoalveolar lavage may be indicated to determine whether there are underlying or predisposing factors for the development or lack of resolution of bacterial pneumonia. 

One limitation of this study is that broader conclusions regarding the use of antimicrobials in dogs with complicated pneumonia cannot be made. It is unknown whether dogs with recurrent pneumonia secondary to aerodigestive disorders such as megaesophagus or local or systemic immune compromise will require a longer duration of antimicrobial therapy. Another limitation of this study is the small sample size in each group likely due to the stringent enrollment criteria. These included the requirement for BAL-confirmed infection and exclusion for prior antimicrobial administration. The main reason for subject exclusion in this study was the previous administration of antimicrobials (52% of cases screened). Without supportive peer-reviewed literature, there is a perception that the collection of BAL is prohibitively risky in dogs with severe respiratory disease. Airway lavage has tremendous value in guiding therapy for dogs with bacterial pneumonia as it can confirm or refute bacterial infection, provide sensitivity testing and may help discern other co-morbid conditions. As the clinical presentation of dogs with CAP (i.e., primary viral pneumonia) and AP (i.e., chemical pneumonitis) can be identical with and without bacterial infection, airway lavage should be considered the standard criterion to guide antimicrobial use. Of note, no dog in the current study had BAL-related complications despite having bacterial pneumonia, underscoring that this procedure can be safely performed even in compromised dogs. This is similar to another study in dogs with respiratory disease undergoing advanced diagnostics, including bronchoalveolar lavage in which complications (if present) were transient and did not contribute to mortality [24]. 

The indiscriminate use of antimicrobials in dogs with clinicopathologic features of pneumonia where bacterial infection is possible but not confirmed remains a common practice and contributes to antimicrobial resistance. A recent study evaluating prescribing practices in an emergency setting reported that only 7% of patients receiving treatment for bacterial pneumonia had the confirmation of presence of bacteria and only 2/28 cases complied with antimicrobial guidelines set at that institution [25]. Another study of dogs with aspiration pneumonia conducted at two tertiary referral institutions documented that only 47/429 (11%) of cases had BAL collection [26]. A final case series of dogs with aspiration pneumonia (10 of which did not have a confirmation of bacteria) documented successful resolution without antimicrobials [27]. In this study, 9 out of 14 dogs had respiratory signs on presentation, all of which resolved within 12–36 h. Although prospective studies are needed, this retrospective study highlighted that antimicrobials may not be necessary in dogs with aspiration pneumonia. Similarly, to that in humans, the inappropriate use of antimicrobials in all dogs with aspiration pneumonia could contribute to the development of resistant pathogens. Improved antimicrobial stewardship will require changing the dogma of the empiric and long duration of antimicrobials to the judicious use of confirmed cases of bacterial infection.

## 5. Conclusions

This pilot study provides evidence to support the reduction in antimicrobial duration from 3 to 6 weeks to a 10-day course in dogs with uncomplicated bacterial pneumonia demonstrating a clinical and hematologic cure. While thoracic radiography has traditionally been a key metric used to determine whether a longer course of antimicrobials is necessary, this study demonstrated that radiographic lesions were commonly persistent despite clinical and hematologic cure. Future studies will be required to best determine the optimal therapy and monitoring in dogs lacking resolution of clinical signs and systemic inflammation within 10 days of therapy initiation and in dogs with complicated bacterial pneumonia.

## Figures and Tables

**Figure 1 animals-11-03096-f001:**
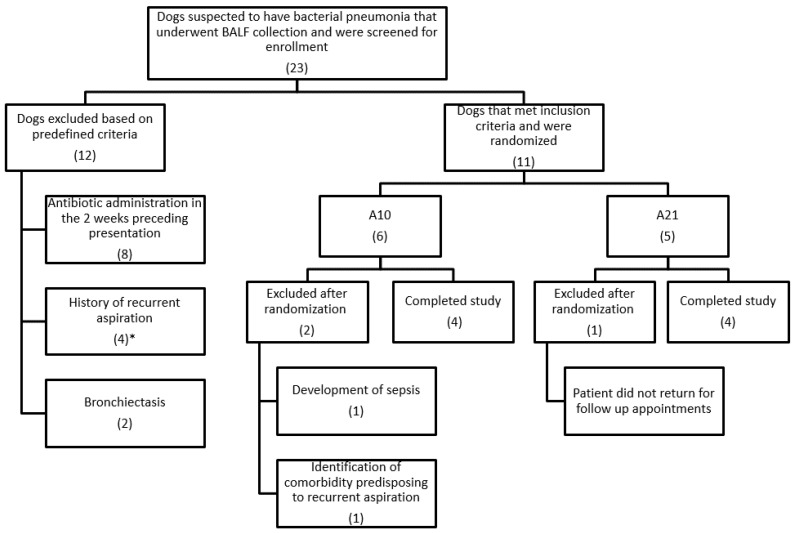
Flow chart indicating patient selection for study enrollment. Numbers in parentheses indicate the number of dogs. A10, patients received 10 days of antibiotics and 11 days of placebo; A21, patients received 21 days of antibiotics; BALF, bronchoalveolar lavage fluid. * Two out of four dogs with historical recurrent aspiration also received antibiotics in the 2 weeks preceding presentation.

**Figure 2 animals-11-03096-f002:**
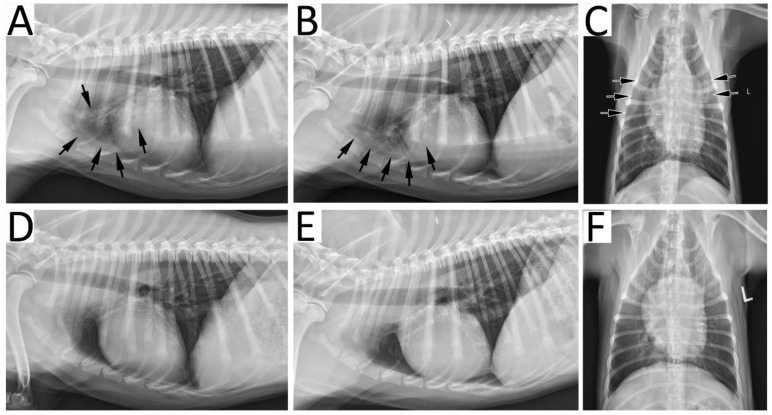
Radiographic resolution of bacterial pneumonia in a 9-month-old male neutered Golden Retriever dog. At initial presentation, right lateral (**A**), left lateral (**B**) and ventrodorsal (**C**) recumbent radiographic projections demonstrated an alveolar pattern (arrows) in the cranial lung fields characterized by increased opacification, air bronchograms and indistinct pulmonary vasculature. Since both the cranial segment of the left cranial lung lobe and right cranial lung lobe were affected, a global radiographic score of 4 was assigned, corresponding to multifocal consolidation. After a full course of therapy, right lateral (**D**), left lateral (**E**) and ventrodorsal (**F**) recumbent projections show resolution of lung lesions and thus, a global radiographic score of 1 was given.

**Table 1 animals-11-03096-t001:** Global radiographic score (GRS) according to a severity scoring system used for the assessment of pneumonia in children and adapted from Taylor et al. [15]. Lung consolidation is defined herein as an area of increased radiographic opacity of the lung in which the vessels are indistinct with or without air bronchograms or lobar margination.

Global Radiographic Score	
Description	Score
Normal	1
Patchy unstructured interstitial pattern and/or bronchial wall thickening without lung consolidation	2
Focal alveolar consolidation but involving no more than one segment or one lobe	3
Multifocal consolidation (specify how many lobes are affected)	4
Diffuse alveolar consolidation	5

**Table 2 animals-11-03096-t002:** Summary of the clinical signs, diagnostic results and final diagnoses for dogs enrolled in each group.

		Clinical Signs/PE Findings						BALF		Final Diagnosis
Group	Dog	Cough	Tachypnea	Respiratory distress	Fever	Duration of signs (d)	Global radiographic score (out of 5)	Septic suppurative inflammation	Culture results	
A10	1	Y	Y	N	Y	2	4	Y ^a^	No growth	CAP
	2	Y	Y	Y	N	<1	4	Y ^a^	*Streptococcus* spp. *alpha hemolytic*	CAP
	3	Y	N	N	Y	5	3	N	*Haemophilus haemoglobinophilus*	Suspected viral pneumonia with secondary bacterial infection
	4	Y	N	Y	Y	1	4	Y ^a^	*Neisseria weaveri* *Ursidibacter maritimus*	AP
A21	5	Y	Y	N	N	2	2	Y ^a^	*Staphylococcus* ^b^ *pasteuri* *Staphylococcus epidermidis* *Staphylococcus saphrophyticus*	AP (post-esophageal foreign body)
	6	Y	Y	Y	N	<1	4	Y ^a^	No growth	Suspected NCPE with VAP
	7	Y	N	N	Y	10	3	Y	*Pseudomonas putida* *Stenotrophomonas maltophilia* *achromobacter xylosoxidans*	Suspected viral pneumonia with secondary bacterial component
	8	Y	Y	Y	N	<1	4	Y	*Pseudomonas aeruginosa* *Streptococcus pluranimalium*	CAP

^a^ Intracellular bacteria present; ^b^ organisms identified in this culture were suspected to be skin contaminants; A10, patients received 10 days of antimicrobial; A21, patients received 21 days of antimicrobials; PE, physical exam; BALF, bronchoalveolar lavage, Y, yes; N, no; CAP, community-acquired pneumonia; AP, aspiration pneumonia; NCPE, non-cardiogenic pulmonary edema; VAP, ventilator-associated pneumonia.

**Table 3 animals-11-03096-t003:** Summary of treatments implemented while hospitalized and after discharge for all patients.

		Treatments Implemented					Antimicrobial Therapy Duration		
Group	Dog	Supplemental O2	IV fluid therapy	Amoxicillin/clavulanic acid (mg/kg)	Enrofloxacin (mg/kg)	Ampicillin sulbactam (mg/kg)	Amoxicillin/clavulanic acid	Enrofloxacin	Time in hospital (days)
A10	1	N	Y	18.5	15	N/A	Administered until day 10	DC after 5 days	2
	2	Y	Y	19.5	15.6	N/A	Administered until day 10	DC after 5 days	4
	3	N	N	18.8	17	N/A	Administered until day 10	DC after 5 days	0
	4	Y	Y	17.5	15	30	DC after 4 days	Administered until day 10	3
A21	5 ^a^	Y	Y	20	15	N/A	Administered until day 21	DC after 6 days	7
	6 ^b^	Y	Y	N/A	20	30	DC after 5 days	Administered until day 21	6
	7	N	Y	17.3	16	N/A	Both antimicrobials DC after 4 days based on susceptibility and replaced with TMS (16 mg/kg PO BID × 7 days)		3
	8	Y	Y	20	12	N/A	DC after 6 days	Administered until day 21	3

^a^ Received one dose of cefazolin 20 mg/kg IV prior to enrollment; ^b^ required mechanical ventilator support from days 2–5 of hospitalization; Y, yes; N, no; N/A, not administered; DC, discontinued after sensitivity results became available.

**Table 4 animals-11-03096-t004:** Time course of the resolution or persistence of clinical signs, hematologic abnormalities and radiographic lesions in 8 dogs receiving short or long courses of antibiotics (*n* = 4/group). Outcome parameters are reported in each group at presentation and follow-up visits. A10, patients received 10 days of antibiotics; A21, patients received 21 days of antibiotics; GRS, global radiographic score.

Outcome	Parameters	Presentation		Follow-Up Evaluations					
				#1		#2		#3	
		A10	A21	A10	A21	A10	A21	A10	A21
Number out of 4 dogs with observed clinical signs	Cough	4	4	1	1 ^a^	1 ^a^	1	1 ^a,b,c^	1 ^a,b,d^
	Tachypnea	2	3	0	0	0	0	0	0
	Respiratory distress	1	2	0	0	0	0	0	0
	Fever	3	1	0	0	0	0	0	0
Number out of 4 dogs with hematologic abnormalities	Segmented neutrophilia	4	2	2 ^e^	1	1	1 ^e^	1 ^e^	0
	Band neutrophilia	0	2	0	1	0	0	0	0
	Neutrophil toxicity	0	3	0	0	0	0	0	0
Radiographic resolution (Score out a maximum score of 5) ^f^	Median GRS per time point	4.0	3.5	1.5	2.5	3.0	1.5	1.5	1.0

^a^ Dog’s cough was characterized as mild, dry and intermittent; ^b^ dogs had a persistent cough, believed to be due to co-morbid conditions, and clinical picture was not consistent with pneumonia; ^c^ dog also had tracheal collapse; ^d^ dog also had laryngeal paralysis; ^e^ one of these dogs had a segmented neutrophilia accompanied by a mild lymphopenia, consistent with a stress leukogram; ^f^ refer to Materials and Methods for description of global radiographic score.

## Data Availability

All the data generated for this study are included in this article.

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
