# Peer review of "Comparison of Short- versus Long-Course Antimicrobial Therapy of Uncomplicated Bacterial Pneumonia in Dogs: A Double-Blinded, Placebo-Controlled Pilot Study"

_animals, 2021, doi:10.3390/ani11113096_

Round 1

Reviewer 1 Report

  1. Overall the study seems an interesting work with significant importance for pet clinicians. However, the sample size is too little to be used in the study.
  2. Most of the literature is older than 5 years, try to find the most recent work and use it to cite in the article.
  3. Table 3: In order to stand-alone abbreviations like N, Y, DC etc should be defined in table footnote.
  4. I suggest subdividing the M&M into parts along with subheadings for better thought of the reader.

Reviewer 2 Report

The manuscript entitled  “Comparison of Short Versus Long Course Antimicrobial  Therapy of Uncomplicated Bacterial Pneumonia in Dogs: a  Double-Blinded, Placebo-Controlled Pilot Study” is a clinical pilot study to evaluate if 10-day course of antimicrobial therapy is sufficient for treatment of uncomplicated canine bacterial pneumonia.

Please find my comments;

Figure 1 the number of dogs excluded from the study was 12, 8 was excluded because of antibiotics administration, 4 history of recurrent aspiration and 2 from bronchiectasis, the total number is 14. Were they the same dogs? The  small number of dogs enrolled to the study (n=4 for each group) is also disturbing.

Table 2. There are different bacterial species identified in two experimental groups of dogs (A10 and A21). Did it influence the final outcome?

Table 4 The dogs with clinical signs and hematologic abnormalities were the same groups or two different?

Lines 317-319. There were own findings of authors or cited from other study?

Lines 352-356; the sentence is unclear

Round 2

Reviewer 2 Report

I recommend the manuscript in present form for publication.